# Accelerated Endothelialization of Nanofibrous Scaffolds for Biomimetic Cardiovascular Implants

**DOI:** 10.3390/ma15062014

**Published:** 2022-03-09

**Authors:** Claudia Matschegewski, Stefanie Kohse, Jana Markhoff, Michael Teske, Katharina Wulf, Niels Grabow, Klaus-Peter Schmitz, Sabine Illner

**Affiliations:** 1Institute for Implant Technology and Biomaterials e.V., Friedrich-Barnewitz-Straße 4, 18119 Rostock, Germany; claudia.matschegewski@uni-rostock.de (C.M.); schmitz@iib-ev.de (K.-P.S.); 2Institute for Biomedical Engineering, Rostock University Medical Center, Friedrich-Barnewitz-Straße 4, 18119 Rostock, Germany; stefaniekohse@web.de (S.K.); jana.markhoff@uni-rostock.de (J.M.); michael.teske@uni-rostock.de (M.T.); katharina.wulf@uni-rostock.de (K.W.); niels.grabow@uni-rostock.de (N.G.)

**Keywords:** nonwoven, cardiovascular stent, human endothelial cells, biocompatibility, CD31

## Abstract

Nanofiber nonwovens are highly promising to serve as biomimetic scaffolds for pioneering cardiac implants such as drug-eluting stent systems or heart valve prosthetics. For successful implant integration, rapid and homogeneous endothelialization is of utmost importance as it forms a hemocompatible surface. This study aims at physicochemical and biological evaluation of various electrospun polymer scaffolds, made of FDA approved medical-grade plastics. Human endothelial cells (EA.hy926) were examined for cell attachment, morphology, viability, as well as actin and PECAM 1 expression. The appraisal of the untreated poly-L-lactide (PLLA L210), poly-ε-caprolactone (PCL) and polyamide-6 (PA-6) nonwovens shows that the hydrophilicity (water contact angle > 80°) and surface free energy (<60 mN/m) is mostly insufficient for rapid cell colonization. Therefore, modification of the surface tension of nonpolar polymer scaffolds by plasma energy was initiated, leading to more than 60% increased wettability and improved colonization. Additionally, NH_3_-plasma surface functionalization resulted in a more physiological localization of cell–cell contact markers, promoting endothelialization on all polymeric surfaces, while fiber diameter remained unaltered. Our data indicates that hydrophobic nonwovens are often insufficient to mimic the native extracellular matrix but also that they can be easily adapted by targeted post-processing steps such as plasma treatment. The results achieved increase the understanding of cell–implant interactions of nanostructured polymer-based biomaterial surfaces in blood contact while also advocating for plasma technology to increase the surface energy of nonpolar biostable, as well as biodegradable polymer scaffolds. Thus, we highlight the potential of plasma-activated electrospun polymer scaffolds for the development of advanced cardiac implants.

## 1. Introduction

Structural and valvular heart diseases are of increasing incidence and a leading cause of mortality worldwide [1,2,3]. Minimally invasive surgery for coronary revascularization by percutaneous coronary intervention (PCI) based on stent implantation has become the standard procedure of care to relieve the stenosis and to restore the dysfunctional stenotic vessel. However, stent surgery is accompanied with local vessel injury, comprising disruption of the intimal smooth muscle cell layer and the luminal endothelial cell layer. The most important clinical complications following PCI are stent restenosis and stent thrombosis, associated with hyperplasia, delayed endothelialization, and acute and chronic inflammation events [4]. Since stent restenosis is mainly related to bare metal stents, drug-eluting stents (DES) have been shown to effectively reduce hyperplasia by comprising a polymer coating with incorporated antiproliferative drugs that inhibit smooth muscle cell proliferation following implantation [5]. However, clinical follow-up studies reported higher rates of late stent thrombosis attributed to DES compared to bare metal stents [6]. Although those DES-related complications are relatively rare, their severity demands further improvements in cardiac implant technology [7].

Contemporary efforts in stent technology focus on polymer-based bioresorbable coronary scaffolds or nanofibrous covered stent systems such as the commercial Papyrus (Biotronik), Jostent Graft-Master (Abbott) or Bioweb (Zeus), in order to decrease incidence of late stent thrombosis [3,8,9,10]. Artificial scaffolds can not only be applied in DES systems but could also be adopted for the development of a variety of other minimally invasive cardiac implants, such as biomimetic heart valves or occluder devices [11]. At the moment, these prosthetics are most commonly xenografts of porcine aortic valves or calf pericardium, with the disadvantages of structural valve deterioration often being accompanied by subsequent valve thickening and calcification, as well as their limited durability [12].

Polymer-based nano- and microfibrous scaffolds hold great potential for overcoming some of the major disadvantages of current polymer stent and cardiac implant designs. Polymeric fibrous scaffolds can be produced in layers or as a composite, in the form of single, core-sheath, blended or co-polymer film or fiber coating systems. Within a comprehensive search using the search engine Web of science, PubMed and Scopus, the increasing interests in electrospun nanofibers over the last 20 years are clearly visible (see Appendix A Appendix A). In addition to synthetic polymers, natural biopolymers can also be used for scaffold fabrication. Intensive research is being conducted by mixing synthetic biopolymers with, e.g., hyaluronic acid derivatives, silk fibroin, cellulose, chitosan, shellac, gelatin, alginates or heparin [13,14,15,16]. The advantages of natural biopolymers can include biostatic properties and biomimetic properties due to extraction from natural sources such as bones, plants, skins, cocoons or algae. However, their source can also be a disadvantage because of their batch and origin dependence. It is difficult to produce reproducible and high-quality implants for medical technology if there is a possibility that (i) allergens are present in the material or (ii) not all components are known or (iii) complex matrices can only be analyzed with great effort.

In contrast, synthetic scaffolds are often limited due to suboptimal biomechanical properties, low biocompatibility or their biodegradation behavior [11,13,14,15]. Since scaffold structure directly influences these properties, the introduction of innovative nano- and microfabrication techniques, such as micropatterning, electrospinning or 3D-bioprinting, will support the development of novel scaffold designs for cardiac implants and similarly will enable the fine-tuning of the desired mechanical requirements as well as biological properties by controlling fiber structure. In particular, electrospinning enables the fabrication of biomaterial scaffolds with tunable parameters, including fiber diameter and alignment, porosity, pore size, inter-connectivity and mesh thickness, which makes them suitable for a range of applications [16,17,18,19]. In addition, electrospinning allows for the fabrication of fused fiber biomaterial scaffolds in the nano- to micrometer range, with fiber diameters starting at 5 nm [20], including the range of feature sizes known to facilitate cellular contact guidance and directed cellular response [21]. Moreover, electrospun nonwovens can offer small pore sizes, up to 90% porosity and pronounced interfibrous integrity [16] with controlled degradation kinetics, if required. Since implants holding the capability of inducing directed cellular response are in high demand, nonwovens are favored, due to their biomimetic surface, which allows for cell attachment, while additionally being highly porous for nutrient transport and cell migration events [19]. Electrospun polymeric scaffolds have already been reported to possess superior capacity in shaping cell morphology, guiding cell migration and affecting cell differentiation in vivo and in vitro [22,23,24,25,26]. The influence of nanofibrous polymer matrices on cell physiology and endothelialization is determined by the specific surface topography and by the physicochemical properties [21,27]. The attachment of cells to the biomaterial surface and the initiated subsequent cell–biomaterial interaction triggers a signaling cascade, which substantially regulates diverse cell functions, including viability, proliferation and the expression of cell-specific proteins [28]. Thereby, alterations in cellular function are suggested to coincide with morphological changes and vice versa.

Alterations in surface topography, e.g., porosity, surface roughness or fiber diameter of electrospun scaffolds have been shown to enhance endothelialization of biomaterials [17,18,29,30]. Additionally, modifications of physicochemical properties such as surface functionalization with plasma, accompanied by changes in hydrophilic properties, have also been demonstrated to influence cell physiology [25].

The first use of plasma-generated amino groups on polymer surfaces to improve blood compatibility was reported in 1969 [31]. Since then, several studies have been performed that have achieved biocompatible surfaces based on the controlled surface properties of the materials, in particular, wettability, chemistry, morphology, crystallinity or surface charge. Plasma modifications, especially cold plasmas, only affect the surface and do not affect bulk properties such as the mechanical and optical properties of the material. Cold plasma is very gentle and therefore suitable for a diverse range of polymers, especially temperature-sensitive materials. Screenings have to be performed for each material and plasma system to optimize parameters to the needs of the desired application. Additionally, a wide range of surface modifications are possible based on the choice of gas or gas mixture. The method can be performed quickly and is suitable for complex geometric samples. In addition to cold low vacuum methods, cold atmospheric plasma systems were developed leading to new applications in plasma medicine such as plasma pens [16,31].

The desired application of the scaffold material, and thus the cell type chosen, is crucial as the same surfaces can show different effects, depending on the cell type, or improving permeation of cell nutrients through nonwovens [32]. For example, enhanced osteogenic differentiation was observed for human mesenchymal stem cells on ammonia plasma treated titanium [33]. Additionally, reduced antimicrobial activity was observed for wound dressings [16,33] or increased C_2_C_12_ proliferation for PLA surfaces [34]. Several studies have been devoted to enhancing the hydrophilicity of polymer surfaces to promote protein and cell attachment [35,36,37,38]. Moreover, plasma surface treatment was shown to directly affect biocompatibility since surface functionalization with NH_3_- or O_2_-plasma has been observed to improve cell growth [39,40,41,42,43,44,45]. NH_3_-plasma generates amino groups on the surface, and it has been demonstrated to be more effective than O_2_-plasma treatments regarding improved cell growth patterns on polymer surfaces or scaffolds [39,42,44,46,47,48,49]. In this context, plasma treatment has been extensively studied and applied to thin films and bulk polymer scaffolds but has not yet been sufficiently investigated for electrospun hydrophobic fiber scaffolds [50]. We selected ammonia plasma based on previous studies of the materials in the field of cardiovascular application [42,49,51]. Thus, this study examined the potential of NH_3_-plasma functionalization of electrospun fibrous biomaterials for promoting cell attachment and physiological growth patterns of human endothelial cells in vitro for the development of biocompatible and biomimetic cardiac scaffolds.

Therefore, we aimed to combine the possibility of electrospinning and low-pressure cold plasma, to meet the needs of medical applications based on the combination of (i) the morphology of nonwoven fibers, with their structure and mechanical properties; and (ii) the surface energy and chemistry of plasma treatment without affecting the fiber morphology. Ammonia plasma treatment was tested on diverse nonwovens of FDA-approved and clinically tested polymers, namely, polylactide (PLA), polycaprolactone (PCL) and polyamide (PA), which represent both biodegradable and biostable biocompatible materials (see Figure 1).

Untreated and plasma-activated polymeric nonwovens were analyzed regarding their influence on endothelial cell characteristics, including viability, spreading and PECAM-1 expression, for judging endothelial activation and phenotype maintenance to evaluate their potential as innovative biomaterials (see Figure 2). These results are a decisive step in the development of novel, advanced scaffolds for cardiac regeneration and a better understanding of cell–biomaterial interactions, not least because endothelialization is a prerequisite for the successful integration of cardiovascular devices, as it forms a natural hemocompatible surface that prevents inflammation and thrombosis events, thus ensuring implant integrity.

## 2. Materials and Methods

### 2.1. Fabrication of Polymeric Nanofiber Nonwovens by Electrospinning

Nanofibrous nonwovens were fabricated out of poly-(L-lactide) (PLLA L210), polyamide (PA-6) and poly-ε-caprolactone (PCL) by electrospinning as following: clear and homogenous polymer solutions of 11 to 12 wt% poly-ε-caprolactone (Capa 6800, Perstop UK Limited) and of 12 wt% polyamide (Ultramid B24N03, BASF, Ludwigshafen, Germany) were obtained by dissolving the polymer in a solvent mixture of formic acid and acetic acid (ratio 1:2, *v*/*v*) at 37 °C. A polymer solution of 2 wt% poly-(L-lactide) (RESOMER^®^ L210, Mw ~400,000 g/mol, Evonik, Essen, Germany) with addition of 1vol% surfactant Triton X-100 (Sigma-Aldrich, Darmstadt, Germany) was obtained by dissolving in a solvent mixture of chloroform and methanol (ratio 4:1, *v*/*v*) at 37 °C.

Fibrous nonwovens were fabricated from these different polymer solutions by free-surface, needleless electrospinning via the Nanospider Lab 200 (ELMARCO, Liberec, Czech Republic) using a rotating wire emitter in a high-volume spinning tube, and a static collector. Emitter-to-collector distances of 18 cm at 16 rpm, 17 cm at 12 rpm or 16.5 cm at 13 rpm were used accordingly for PLLA L210, PCL and PA-6, each resulting in nonwoven samples with randomized fibers. The applied high voltages were 49 to 58 kV, 60 to 80 kV and 72 to 76 kV for PLLA L210, PCL and PA-6, respectively, each under ambient conditions of 23 °C and humidity of 35%. The generated polymeric nonwoven mats were dried for 12 h at 40 °C using the vacuum oven VO 200 (Memmert GmbH and Co., Schwabach, Germany, 40 mbar). 

### 2.2. Nonwoven Characterization by Scanning Electron Microscopy

Fiber morphology of the PLLA L210, PCL and PA-6 nonwovens was examined by scanning electron microscopy (SEM) using a QUANTA FEG 250 (FEI Company, Dreieich, Germany) with an Everhart–Thornley secondary electron detector (ETD) at an acceleration voltage of 10 kV; a working distance of around 10 mm, at a high vacuum of 3.5 × 10^−6^ mbar; and a spot size of 3.0. The samples were fixed onto aluminum trays with conductive tape and sputter coated with gold by Agar Sputter Coater (Agar Scientific Ltd., Essex, UK). Therefore, the samples were put under vacuum of 0.2 mbar and exposed to gold flow 2 times for 120 s. SEM images were taken at magnification 500×, 1000×, 5000× and 8000×. For quality assurance of the produced polymeric nonwovens and for determining the average fiber diameter, SEM analysis was performed at different areas of the nonwovens. Fiber diameters were calculated from SEM images by using EDAX Genesis software, measuring 50 random fibers from five micrographs for each nonwoven at high magnification.

### 2.3. Surface Plasma Modification of Nonwovens

Plasma-chemical surface modification was conducted by plasma etching (PE) of nonwoven samples in an ammonia (NH_3_) plasma, generating radical species and amino groups. The short plasma activation process was performed for 1 min and 60% generator output in an ammonia radio frequency (RF) plasma generator (frequency 13.56 MHz, power 100 W, Diener electronic GmbH and Co. KG, Ebhausen, Germany) at a low pressure of 0.3 mbar based on previously studies [42,49,51]. The screening data were not presented because the energy density of the plasma in the chamber depends on a wide range of factors such as chamber material, design, sample positioning and mounting, electrode spacing, size, shape and material, and the method of excitation. Generalization or the transfer of parameters from one system to another is not possible, so screening of suitable parameters depending on the objective must always be performed.

### 2.4. Water Contact Angle and Surface Free Energy

Water contact angle measurements were performed by the sessile drop method (water) on the nonwoven polymer surface using a goniometer (OCA 20, Dataphysics Instruments GmbH, Filderstadt, Germany) equipped with SPSS software 15.0. Nonwovens of PLLA L210 were washed three times for 10 min each with pure water to remove Triton X-100. Nonwovens were attached to glass slides, and water contact angles were determined by the sessile drop method with water droplets of 5 µL. A time-resolved measurement over 60 s was performed, whereby the smallest standard deviation was obtained after 10 s. Mean values and standard deviations were calculated from five independent samples with *n* = 4 measurements per sample.

To calculate the surface free energy (SFE) of untreated nonwovens according to Owens–Wendt–Rabel–Kaelble (OWRK) [52], further measurements were performed with a mobile surface analyzer (MSA, KRÜSS GmbH, Hamburg, Germany) with ADVANCE 1.9.2 software. The initial contact angles of two liquids, water and diiodomethane, were determined against air, whereby drops with a volume of 1 or 2 µL were deposited and measured within a few seconds (*n* = 3).

### 2.5. Cell Culture

Human vascular endothelial cells EA.hy926 (ATCC^®^, CRL-2922™, Manassas, VA, USA) were cultured in Dulbecco’s Modified Eagle Medium (DMEM), including 4.5 g/L glucose and 3.7 g/L NaHCO_3_ with 10% fetal calf serum (FCS) (both: PAN Biotech, Aidenbach, Germany) and penicillin (100 units/mL), streptomycin (100 ng/mL) at 37 °C, and 5% CO_2_ under humidified atmosphere. For the experiments, cells were seeded at a concentration of 2 × 10^4^ cells/cm^2^ on the nanofibrous polymer scaffolds and incubated for 48 h at 37 °C and 5% CO_2_ under humidified atmosphere.

### 2.6. Cell Viability Assay

Cell viability of human endothelial EA.hy926 cells was determined using the CellQuanti-Blue™ assay (BioAssaySystems, Hayward, CA, USA) according to the manufacturer’s instructions. Briefly, cell viability was assessed via quantification of cellular metabolic activity by the reduction of the substrate resazurin to resorufin by cellular reductases. Therefore, cells were incubated with CellQuanti-Blue™ as 10% of the culture medium volume for 2 h after a 46-h cell cultivation period on the polymeric surfaces. The resulting fluorescence of resorufin was measured at an emission wavelength of 590 nm with an excitation wavelength of 544 nm using a microplate reader (FLUOstar OPTIMA, BMG Labtech, Offenburg, Germany). For each polymer, six independent biological replicates were measured. Data were normalized to viability of EA.hy926 cells grown on planar non-cytotoxic tissue culture polystyrene (TCPS) as control surface (NC) [40,42].

### 2.7. Cell Morphology Analysis by Scanning Electron Microscopy

Cell morphology of human endothelial EA.hy926 cells grown for 48 h on either polymeric nonwovens or tissue culture polystyrene control surface (NC) was observed by scanning electron microscopy. After the incubation on the polymeric surfaces, cells were fixed with 2.5% glutaraldehyde and 0.2 M sodium cacodylate, in PBS for 30 min. Samples were then washed with sodium phosphate buffer, dehydrated in a graded series of ethanol (50%, 75%, 90% and 100%) and dried with CO_2_ in a critical point dryer (CPD 7501, Quorum Technologies Ltd., Laughton, Lewes, East Sussex, UK). Samples were sputter-coated with gold by Agar Sputter Coater (Canemco Inc., QC, Canada), and image acquisition was performed with the scanning electron microscope Quanta^TM^ FEG 250 (FEI Company, Hillsboro, OR, USA) at 10 kV under high vacuum conditions by using the Everhart–Thornley secondary electron detector (ETD).

### 2.8. Endothelialization Analysis by Cell Spreading and Cell Shape Index

Endothelialization potential of polymeric nonwovens was assessed by quantification of cellular spreading of EA.hy926 endothelial cells and analysis of phenotype maintenance, evaluated by cell shape index after a 48 h cultivation period on all surfaces. This quantification was done based on SEM images of fixed cells. For determination of cell spreading, cell areas of 40 cells per specimen of three independent replicates were measured using the area measurement function of ImageJ software. Cell shape index (CSI) analysis was performed with ImageJ software by using the formula CSI = 4π × area/(perimeter)^2^ [53]. Calculated CSI defines cellular morphological shape ranging from 0 to 1, corresponding to a circular shape (CSI = 1) or a straight line, i.e., maximum elongated shape (CSI = 0).

### 2.9. Immunofluorescence of PECAM-1 (CD31) and Actin Cytoskeleton

For immunostaining, endothelial EA.hy926 cells were examined after incubation for 48 h on the polymeric nonwovens as well as on the control surface (NC) at 37 °C and 5% CO_2_ under humidified atmosphere. Cells were fixed in 4% paraformaldehyde (PFA) for 30 min at room temperature (RT), rinsed in PBS (pH = 7.4) and permeabilized with Triton X-100 (Sigma-Aldrich, Darmstadt, Germany) for 30 min at RT. Cells were then incubated with primary murine monoclonal anti-human PECAM-1 (CD31) antibody (1:20, DAKO, Agilent, Santa Clara, CA, USA) overnight. Afterwards, cells were rinsed in PBS and incubated with secondary donkey anti-mouse antibody conjugated with Alexa Fluor 488 (Life Technologies GmbH, Darmstadt, Germany) for 1 h at RT. For actin staining, TRITC-conjugated phalloidin (500 μg/mL, Sigma-Aldrich, Taufkirchen, Germany) was used by incubating the cells in the staining solution for 1 h at RT. Cell nuclei were stained with Hoechst 33,342 (1:500, Sigma-Aldrich, Taufkirchen, Germany) for 1 h at RT. Cells were mounted in VectaShield mounting medium (Vector Laboratories, Burlingame, CA, USA) and examined by confocal laser scanning microscopy (FluoView FV1000, Olympus, Hamburg, Germany). Quantification of mean fluorescence intensity of the respective markers in cell images was evaluated by CellProfiler software.

### 2.10. Statistical Analysis

Data were reported as mean value with standard deviation and analyzed by one-way ANOVA carried out with GraphPad©Prism 5 software (La Jolla, CA, USA). Statistical significance was defined as ns = not significant *** *p* < 0.001.

## 3. Results

### 3.1. Morphology of Polymeric Nonwovens

Different polymeric nonwovens were produced via electrospinning and then compared in terms of fiber diameter. Representative images of the three polymer classes are shown in Figure 3 (additional magnifications of the SEM images are presented in Appendix A). Morphological differences between the individual polymer classes are evident but not after NH_3_-plasma treatment. The PLLA L210 fibers are about twice as thick as the PCL fibers and four times thicker than PA-6 fibers.

The fiber diameters of the different polymeric nonwovens were determined from 50 measurements each and are shown in absolute terms in Figure 4. Whereas the diversity of the fibers varies the most within PLLA nonwovens, the fiber diameter of untreated and plasma functionalized PLLA L210, PCL and PA-6 nonwovens differ only marginally. This implies that the fiber structure is not changed by the plasma treatment. Additionally, the frequency of the respective fiber diameters for each investigated polymer is presented in Appendix A.

### 3.2. Wettability Analysis and Surface Free Energy of Polymeric Nonwovens

First, the wettability of the nonwovens was determined in a time-resolved manner (Figure 5A). A second measurement, performed instantaneously after only a few seconds, provides a direct comparison of the initial wettability of untreated and plasma-activated nonwovens (Figure 5B).

It can be seen that the water contact angle (WCA) of PA-6 nonwovens does not remain constant and decreases rapidly over 60 s (Figure 5A). In contrast, the WCA of the PLLA L210 and PCL nonwovens is very high at 130 to 140° as expected, which is due to the special surface morphology of nonwoven structures. As opposed to film or foil materials, nanofibrous nonwovens have pores that are filled with air. The wetting properties of nonwovens are influenced by their weight, layer thickness or (more precisely) their pore structure. Such a surface, where air is trapped between the liquid and the solid and thus can only be wetted incompletely, is called a composite interface. [54] The initial WCA before and after plasma treatment is shown in Figure 5B. It can be clearly seen that the WCA for all investigated nonwovens can be greatly reduced by one-minute NH_3_-plasma treatment, indicating that plasma treatment is very efficient at improving the wettability of nonwoven fibers.

Further results on surface free energy and its division into polar and disperse fractions are summarized in Table 1. For their calculation by means of OWRK, the diiodomethane contact angle was additionally determined. The surface free energies of untreated PLLA L210 and PCL nonwovens differ only marginally. The SFE of PA-6 nonwoven is slightly lower, but in this case the measurement is affected by a high standard deviation. After plasma activation, the SFE is noticeably improved for PCL and PA-6, whereby the disperse fraction is reduced and the polar fraction strongly increased for all polymers.

### 3.3. Biocompatibility of Polymeric Nonwovens Assessed by Cell Viability

Cell viability of human endothelial EA.hy926 cells grown for 48 h on untreated and NH_3_-plasma modified polymeric nanofiber nonwovens of PLLA L210, PCL and PA-6 is shown in Figure 6. Values were normalized to values from the control surface (NC, TCPS), which was set to 100%. For the untreated nonwovens, relative cell viability was highest for PLLA L210 (57.1%), followed by nonwovens of PA-6 (54.1%) and PCL (49.6%); however, these differences were not statistically significant.

In contrast, NH_3_-plasma modification has indeed shown to increase the relative cell viability of endothelial EA.hy926 cells on all polymeric nonwovens. This effect was most prominent for PCL nonwoven, where NH_3_-plasma modification yielded the highest and most statistically significant (*p* < 0.001) increase in EA.hy926 cell viability of 75.6%, corresponding to an enhancement of 26.0% compared to untreated PCL nonwovens. Additionally, for PLLA L210 and PA-6 nonwovens, a positive effect of NH_3_-plasma treatment on cell viability was observed, although it was not significant and slightly lower than what was detected for PCL. NH_3_-surface treatment of PLLA L210 and PA-6 resulted in an increased viability of endothelial EA.hy926 cells up to 61.3% for PLLA L210 and 63.0% for PA-6.

### 3.4. Influence of Nanofiber Scaffolds on Cell Attachment and Morphology

Cell morphology analysis using scanning electron microscopy revealed phenotypic differences of human endothelial EA.hy926 cells grown on untreated nonwovens of PLLA L210, PCL and PA-6 compared to those grown on NH_3_-plasma-modified polymeric nonwovens (Figure 7). Only moderate cell attachment and spreading of EA.hy926 cells could be observed on the untreated nanofiber nonwovens of PLLA L210, PCL and PA-6. In particular, cells that were grown on electrospun PLLA L210 and PCL scaffolds were more likely to exhibit spherical phenotypes than cells that were grown on PA-6 nonwovens where they appeared more flattened. Especially on PLLA L210 nonwovens, EA.hy926 cells were shown to grow around the scaffold nanofibers to some degree. After NH_3_-plasma treatment, all of the three types of polymeric nonwovens were shown to facilitate better cell attachment of human endothelial cells and showed increased cell spreading of EA.hy926 cells when compared to the corresponding untreated polymer surfaces. In general, endothelial EA.hy926 cells grown on all NH_3_-plasma treated surfaces were larger and exhibited a flattened and more elongated phenotype with more filopods than on the unmodified nonwovens. Furthermore, this altered effect on cell spreading and the phenotype change was most obvious for cells that were grown on NH_3_-plasma treated PLLA L210 nonwovens. It was also apparent that on all nanofibrous scaffolds, human endothelial cells were able to spread between individual fibers of the mats, but none of the cells were observed to have grown deeper into the pores of the analyzed nonwovens.

Regarding phenotype maintenance, the morphology of human EA.hy926 cells grown on the NH_3_-plasma modified polymeric nonwovens was comparable to those on the control surface (NC), although on the control surface, cells generally exhibited a much larger cell area compared to those on all of the polymeric nonwovens.

### 3.5. Quantification of Endothelialization

To evaluate the impact of NH_3_-plasma treatment on endothelialization potential as a substantial biological requisite for successful cardiac scaffolds, cell spreading of endothelial EA.hy926 cells was quantified by measuring cell areas (Figure 8). Measurements were therefore conducted on cells either grown on NH_3_-modified or untreated polymeric nonwovens of PLLA L210, PCL and PA-6.

Mean cell area of human endothelial EA.hy926 cells on untreated nonwovens ranged from 144.9 µm^2^ to 203.2 µm^2^ for PLLA L210 and PA-6, respectively. Modification with NH_3_-plasma led to a remarkable increase of the cell area of human endothelial cells for all three types of investigated polymeric nonwovens. The effect was most obvious for PLLA L210 and PA-6 nonwovens, where the cell area was significantly increased and even doubled after NH_3_-plasma treatment, ranging from 404.5 µm^2^ to 406.0 µm^2^, respectively. The cell area also tended to increase on PCL nonwovens after NH_3_-plasma modification, although the difference was not statistically significant compared to untreated PCL mats. However, cell area of human EA.hy926 cells was highest on the control surface (NC), ranging from 1565.0 µm^2^ for NC and 1873.0 µm^2^ for NH_3_-plasma treated NC.

In order to judge the quality of endothelialization based on maintenance of the endothelial phenotype, cell shape of human endothelial cells was analyzed by calculating cell circularity by cell shape index (CSI). The index, ranging from 0 to 1, is either corresponding to a circular shape (1) or a straight line (0 for maximum elongated shape). CSI of human endothelial cells on untreated polymeric nonwovens ranged between 0.64 for PCL nonwoven and 0.73 for PLLA nonwoven, compared to 0.61 for the control surface (NC) (Figure 9). After NH_3_-plasma modification, CSI was diminished on all polymeric surfaces, with lower circularity measurements averaged to 0.67, 0.65 and 0.58 for PLLA L210, PCL and PA-6 nonwovens, respectively, thus representing a less circular shape and more elongated phenotype of human EA.hy926 cells on the NH_3_-plasma modified nonwovens. In particular, CSI was lowest on plasma-modified PA-6 nonwoven and therefore expressed the smallest difference compared to control surface with respective CSI of 0.54. Nonetheless, cells were not detected to be aligned in one specific direction.

### 3.6. Endothelial Actin Cytoskeleton Formation on Polymeric Nonwovens

The formation of the actin cytoskeleton of human endothelial EA.hy926 grown on specific polymer nonwovens is shown in Figure 10 (see also Appendix A) with regard to NH_3_-activation status and is compared to the respective control surface (NC).

On all of the unmodified polymeric nonwovens comprising PLLA L210, PCL and PA-6, EA.hy926 cells was shown to exhibit a much reduced actin formation. This reduction in actin formation was evident due to the merely faint background staining of F-actin, with a concentrated F-actin ring at the outer cell membrane and a lack of any any visible intracellular actin fibers. In contrast, intracellular actin fibers could be observed spanning the entire cell body on the control surface.

NH_3_-plasma treatment also showed a striking positive effect on the intracellular cytoskeleton formation regardless of the observed polymeric nonwovens. In particular, intracellular F-actin expression was enhanced on PLLA L210, PCL and PA-6 nonwovens when compared to the respective untreated counterparts. However, distinct stress fiber formation, as it is seen in EA.hy926 cells grown on the control surface, still remained relatively poor on all of the plasma-activated polymer nonwovens.

### 3.7. Expression of Endothelial Cell-Specific PECAM-1 Marker

Immunofluorescent staining was examined to determine the expression of endothelial cell specific cell–cell contact PECAM-1 (CD31) marker in human EA.hy926 endothelial cells growing on polymeric nonwovens of PLLA L210, PCL and PA-6, with or without NH_3_-plasma surface modification and control surface (NC). Analysis of CD31 expression was used as a positive indicator of successful formation of cell–cell junctions in human endothelial EA.hy926 cells. Results indicate that on all untreated polymeric nonwovens, only a sparse CD31 expression in EA.hy926 cells was observed compared to the control surface (NC) (Figure 11 and Appendix A).

On the contrary, EA.hy926 cells that were grown on NH_3_-plasma-modified polymeric nonwovens demonstrated an increased CD31 expression that was concentrated in the cell-to-cell contact sites. Thus, CD31 expression patterns on NH_3_-plasma-modified nonwovens were more similar to general CD31 expression on the respective control surface. While comparing the different NH_3_-plasma-treated nonwovens, CD31 expression was highest and most homogeneous for EA.hy926 cells grown on NH_3_-plasma treated nonwovens of PA-6. Here, it showed a faint intracellular constitutive background expression with intense cell–cell contact sites and thus was most comparable to the control (NC).

## 4. Discussion

Advancements in micro- and nanotechnology provide the technical platform to fabricate innovative biomaterials for tissue engineering in order to restore, maintain or even improve biological function. Electrospun polymeric scaffolds possess similar structural and morphological properties to extracellular matrix that might make them excellent biomimetic scaffolds for several cardiac interventions [18,26,55,56,57,58]. This includes drug-eluting systems, artificial heart valve prosthetics, or occluder systems. Although contemporary efforts have been made in stent technology regarding efficacy and safety, polymeric scaffolds are still facing challenges such as inflammation or late thrombosis events [9]. Delayed endothelialization is believed to play a key role in the occurrence of late stent thrombosis, and therefore research has attempted to improve stent designs to accelerate endothelialization. Since the endothelium represents an inherently antithrombogenic surface, fast formation of an intact endothelium at the implant site can prevent thrombus formation, or inflammation, events.

The present study is focused on the fabrication of nanofibrous polymer matrices as biomimetic scaffolds and the characterization of their physicochemical properties and biological performance in order to evaluate their suitability for the development of innovative artificial grafts for structural and valvular heart diseases [11,15,57]. Additionally, the effect of plasma functionalization of polymeric nonwovens was evaluated in attempt to improve endothelialization and thus biocompatibility of polymeric nanofibrous matrices [42,48,59]. Critical cellular parameters of human endothelial cells were examined to determine distinct growth patterns on polymeric nonwovens and investigate whether plasma functionalization with NH_3_ affects biocompatibility and the maintenance of the endothelial cell phenotype.

### 4.1. Surface Characterization and Chemical Modification by Plasma Treatment

This study demonstrated that nanofibrous polymer scaffolds of biodegradable PLLA L210 and PCL as well as of biostable PA-6 were successfully fabricated by needleless electrospinning. Polymeric nonwovens exhibit uniform meshes with randomly distributed fibers with mean fiber diameters of 450 ± 250 nm for PLLA L210 and 100 ± 50 nm for PA-6 and 200 ± 50 nm for PCL while forming interconnected pores. Additionally, fiber morphology was shown to lack the formation of beads and junctions, which indicates sufficient evaporation of organic solvent mixtures during the electrospinning process.

The physicochemical surface analysis of polymeric nonwovens demonstrated clear differences in wettability and surface free energy among the fabricated polymeric nonwovens (see Table 1). Water contact angles of unmodified polymeric nonwovens ranged from hydrophobic to highly hydrophobic, which likely indicatives that, despite the chemical composition, the hydrophobicity is a consequence of the nanostructured surface topography itself [60,61,62,63]. Contact angle measurements confirmed that surface treatment with NH_3_-plasma increases hydrophilicity of all types of investigated polymeric nonwovens. Moreover, the NH_3_-plasma led to a higher polar fraction of the surface free energy for all examined polymeric nonwovens. Thus, NH_3_-plasma functionalization was observed to exhibit a significant decrease of water contact angles (less than 40°) after NH_3_-plasma deposition for all polymeric nonwovens, indicating a more hydrophilic surface. Among the tested polymer scaffolds, the very high contact angles of untreated PLLA L210 and PCL nonwovens, compared to the nearly zero apparent contact angles after NH_3_-plasma-treatment, might be induced by capillary effects and the highly porous fibrous structure. That effect was observed to be highest for PLLA L210 nonwovens, while PCL- and PA-6 nonwovens exhibited less than half of the mean fiber diameters of PLLA L210.

### 4.2. Evaluation of Cell Physiology and Cell Phenotype Maintenance on Nanofibrous Polymer Scaffolds

Generally, cells are known to be able to sense aspects of their environment, including distinct surface topographical and chemical features, and adopt cellular physiology in response to those physicochemical properties of the biomaterial [28]. Thus, the results of the present study show that the nanofibrous topography of the fabricated electrospun polymeric matrices alone influences cell physiology and phenotype maintenance of human endothelial EA.hy926 cells, while distinct chemical characteristics, due to NH_3_-plasma treatment, have a significant influence on these properties. In this study, human endothelial EA.hy926 cells were successfully proven to grow on fabricated nanofibrous nonwovens of PLLA L210, PCL and PA-6. However, for all of those untreated nonwovens, only a moderate biocompatibility, assessed by cell viability analysis (ranging from 49.6 to 57.1%), could be observed. Correspondingly, compared to the planar control surface, cells grown on the polymeric nonwovens also demonstrated less cell adhesion, as well as reduced cell viability nearly independent of the type of polymer. This could be a consequence of the nanofibrousness of the polymeric scaffolds, causing initially altered cell physiology affecting cell attachment, cell morphology and cell-functional parameters. Beyond the effect of the nanofibrous surface, in general, the nonwoven scaffolds also provide a different growing environment since the nonwovens are a more three-dimensional scaffold compared to the planar negative control (NC), which is simply two-dimensional with an additionally cell culture treated surface [40]. In consequence, the different dimensions between nonwoven and the planar control surface might be the reason for the observed differences in cell attachment and vitality of human endothelial EA.hy926 cells. In this context, Del Gaudio et al. [56] also reported lower cell vitality for primary human endothelial HUVEC cells seeded on PCL nonwovens in comparison to a planar TCPS surface, which is in agreement with the results on cell viability and attachment in the present and prior studies [42].

Regarding the evaluation of distinct cell–nanofiber interactions, Ahmed et al. [55] also described alterations in cell shape and function of primary human endothelial HUVEC cells grown on nano- and microfibrous poly(lactic-co-glycolic acid) PLGA meshes in a fiber diameter dependent manner. In particular, HUVECs became more spherical with smaller fiber diameters in the nanometer range (up to 500 nm), indicated by higher cell shape index, when compared to cells grown on micrometer fibers. Similarly, the results in the present study also demonstrate high CSI values of human endothelial cells on the tested nanofiber polymeric nonwovens.

In addition to the nanofibrous surface topography that might be responsible for distinct growth patterns of endothelial cells on polymeric nonwovens, the observed differences in wettability and surface free energy of the polymeric meshes could also contribute to changes in cell morphology. Limited cell attachment and viability can be attributed to the high hydrophobicity of all nanofibrous polymer surfaces that are known to negatively influence cell growth patterns. According to other studies, material surfaces that possess moderate hydrophilicity, i.e., water contact angle ranging between 40° and 80°, are shown to be favored by several cell types regarding cell attachment and vitality [64,65]. Therefore, the results of this study suggest that enhanced cell adhesion and viability of human endothelial EA.hy926 cells grown on PLLA L210 and PA-6 nonwovens might be attributed to the higher hydrophilicity than of PCL nonwovens. However, it is worth noting that the hydrophilic surface of PLLA L210 (less than 20° before washing, results not shown) was artificially induced by the addition of a surfactant and could not be maintained after the introduction of washing processes. Consequently, after release of Triton X-100, the contact angle of PLLA L210 was highly hydrophobic and similar to that of PCL nonwoven (see Table 1). As a general conclusion, it is essential to consider all additives used during production to simplify the electrospinning process and for a correct comparison of different final polymer fiber surfaces.

While distinct fiber geometries and surface wettability among the investigated untreated polymeric nonwovens were exhibited, differences in cell viability and cell shape were only slight and not significant. This might also be based on only slight differences in fiber diameters within the three polymer scaffolds, which only ranged within 100 to 450 nm for PCL, PA-6 and PLLA L210 nonwovens. In consequence, nanofiber-dependent differences in cell morphology and physiology patterns might become more obvious when comparing a higher range of fiber diameters. This might be evidenced by other studies that reported altered cell physiology according to differences in fiber diameters of polymeric nonwovens. Here, Ko et al. [58] showed increased proliferation and growth behavior of endothelial cells with increased fiber diameters of PLGA matrices by comparing nonwovens produced in nanofiber diameters of 200 nm and 600 nm.

Moreover, since no significant differences regarding cell viability and growth patterns among the investigated polymer nonwovens could be demonstrated, it seems that surface chemistry might be less influential than surface topography because endothelial EA.hy926 cells exhibited similar growth patterns on all polymeric nonwovens, regardless of their distinct chemical composition, i.e., PLLA L210 vs. PCL vs. PA-6. In regard to this differentiation between the relative influence of topographical and chemical effects, Cousin et al. [66] showed that changes in the morphology of fibroblasts could be predominantly attributed to the surface topography of nanoparticulated coatings, irrespective of the surface chemistry, which is consistent with the results of the present study.

### 4.3. Plasma Functionalization of Nanofibrous Polymer Scaffolds towards Improved Biocompatibility and Cellular Growth Patterns

Surface functionalization with NH_3_-plasma was applied in order to improve biocompatibility and endothelialization of the polymeric nonwovens and thus their biological functionality and integrity regarding their usage as potential artificial cardiac or cardiovascular grafts.

Indeed, NH_3_-plasma functionalization resulted in remarkable elevation of biocompatibility and phenotype maintenance of human endothelial cells on all of the investigated nanofibrous polymer scaffolds. In particular, NH_3_-functionalization increased cell viability by up to 26% above untreated nanofibrous nonwovens as observed for PCL scaffolds. Additionally, phenotypic traits of endothelial cells were improved after NH_3_-treatment, seen in enhanced cell spreading along with more elongated cell shapes (represented by lower CSI) as well as improved formation of actin cytoskeleton and more physiological CD31-expression. In consequence, surface functionalization with NH_3_-plasma was proven to exert positive effects on endothelialization, which could potentially improve biocompatibility and hemocompatibility of PLLA L210, PCL and PA-6 nanofiber matrices. Those results are in accordance with other studies that also have shown a positive effect of NH_3_-plasma of polymer films on cell viability and expression of endothelial markers such as CD31 in human endothelial HUVEC and HCAEC cells [42]. The positive effects seen in this study of NH_3_-plasma treatment of the polymeric nonwovens of PLLA L210, PCL and PA-6 could be attributed to the incorporation of functional NH_2_-groups that caused an increase in hydrophilicity of the polymeric nonwovens, subsequently leading to improved cell viability, which is also consistent with previous studies [67]. Because the surface morphology characterization did not reveal any significant effect of plasma treatment on the fiber diameters, it seems plausible that the enhancement of cell spreading and viability on the plasma-treated surfaces was induced solely by the increased hydrophilicity and the incorporation of amino functionalities, i.e., NH_2_-groups. In particular, NH_2_-group surface coupling might facilitate the adsorption of serum proteins to the surface that elevates cell attachment, subsequently leading to improved spreading and improved physiological patterns of human endothelial cells. Additionally, surface-adsorbed serum proteins might possess favorable configurations that are advantageous for cell attachment, spreading and growth patterns, as previous studies already reported [68]. Previous studies that demonstrated increased cell spreading of endothelial cells on NH_3_-treated fibrous PLLA scaffolds postulate that the long-term beneficial effect of NH_3_ plasma-treatment of polymer surfaces is a result of NH_2_-group coupling to the surfaces [42,45].

### 4.4. Limitations and Potential of Plasma-Activated Nonwovens for Cardiovascular Applications

Within this study, we were able to gain basic knowledge about the biological interaction between cell-nonwoven surfaces under the influence of NH_3_ plasma activation. However, so far we have only been able to make limited statements about the colonization of the fiber networks with human endothelial cells [58,69,70]. Both untreated and plasma-treated nonwovens were only studied within in vitro cell culture for short periods of time (48 h). In literature, a doubling time of approximately 25 h of the used endothelial cells is described [71,72]. Therefore, longer cell culture experiments in the future could yield more information. As already mentioned before, no general conclusions about polymer- or nonwoven-specific plasma treatment can be predicted. In general, parameters such as polymer type and electrospinning processing or plasma treatment time need to be adjusted to the desired applications. In addition, insights into biological response such as cell colonization and inflammatory behavior are limited in vitro experiments. In preparation for in vivo experiments, new methods such as cell behavior under flow conditions have to be established to quantify effects of shear stress on a confluent endothelial cell layer and to examine more complex situations such as biofilm formation or thrombotic events.

According to literature [11,69,73], the following important current questions cannot yet be conclusively or adequately addressed: (i) how long does complete endothealization of nonwoven materials take in vivo (with blood circulation)? (ii) do specific cells penetrate through the fiber network and does biodegradation influence the functionality? (iii) which polymer type should be preferred for which implant region—biodegradable vs. biostable?

In addition, Table 2 provides a brief summary of further considerations regarding plasma surface treatment of electrospun polymer scaffolds.

## 5. Conclusions

In this study, nanofibrous scaffolds of PLLA L210, PCL and PA-6 were successfully fabricated by electrospinning for enabling their usage as biomimetic matrices for endothelialization. Results showed surface-topography-dependent alterations in cell physiology and endothelialization potential of human EA.hy926 endothelial cells. Untreated polymer meshes showed moderate biocompatibility and low cell spreading, which might complicate their use as implants. In order to improve the biocompatibility and biological integrity of the nanofibrous polymeric scaffolds, surface modification by NH_3_-plasma functionalization was performed as it is described as a promising method to optimize biomaterials towards better cell compatibility and implant integrity. Chemical surface modification by NH_3_-plasma treatment was demonstrated to alter hydrophilicity of all investigated polymer nonwovens while not affecting fiber morphology or structure integrity. Short functionalization with NH_3_-plasma was demonstrated to effectively promote cell attachment and cell growth patterns of human endothelial cells on polymer nanofiber scaffolds. The positive effect was proven by enhanced cell attachment and spreading as well as increased cell viability. Thus, surface functionalization by NH_2_-plasma could further promote endothelialization and hemocompatibility of polymeric nonwovens of PLLA L210, PCL and PA-6, making them suitable as advanced biomaterials for several cardiac and vascular interventions.

Further studies should more closely examine the influence of fiber thickness and pore size of nanofiber nonwovens using one specific polymer type. We know that, for instance, flat human endothelial cells forming a uniform monolayer with cell clusters of 200–400 µm are probably too large for the pores of our nanofiber meshes [74]. However, complex biodegradable PLLA or PCL matrices, where fiber degradation and cavity generation over time plays a role, need to be considered in vitro. To study degradation or ingrowth behavior under in vivo conditions for longer time periods, fast-degrading medically approved polymers could also become interesting as model fiber systems, if the influence of the polymer type on cell colonization is incidental. In addition, growth-promoting substances can be released via fiber degradation [74].

Controlling the fabrication parameters of the electrospinning process to optimize fiber diameter, pore structure, and mesh density and thickness is an important task to ensure adequate cell functionality and biological integrity. Together with the finely tunable fabrication properties of the electrospinning technique, plasma-treated polymer nonwovens are promising biomaterials for specifically directing cellular response. Thereby, systematic modulation of biomaterial’s physicochemical properties will support the elucidation of key cell–biomaterial interactions to further improve implant technology.

## Figures and Tables

**Figure 1 materials-15-02014-f001:**
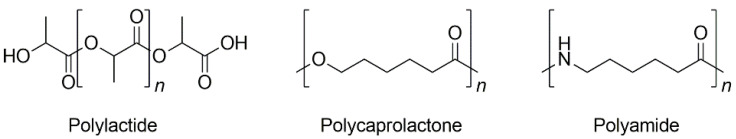
Chemical structure of the well-established polymer types for biomedical application used in this study.

**Figure 2 materials-15-02014-f002:**
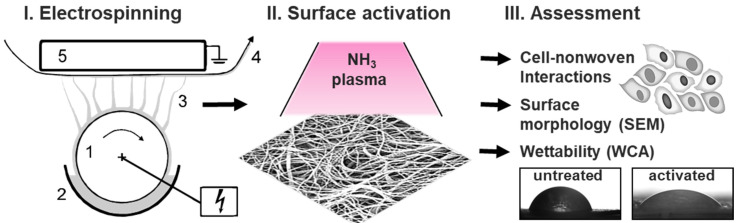
Systematic illustration of the study design and methodological background. (**I**). Preparation of the different used nonwovens using Elmarco nanospider setup, including (1) high-voltage source at the rotating emitter; (2) a bath filled with a polymer solution; (3) fiber formation under solvent evaporation; and (4) a nonwoven collecting unit, (5) which is electrical driven. (**II**). Surface plasma activation of a nonwoven illustrated by a SEM image. (**III**). Characterization of the prepared polymer-based nonwovens via comprehensive biological studies, SEM and contact angle measurements, before and after plasma treatment.

**Figure 3 materials-15-02014-f003:**
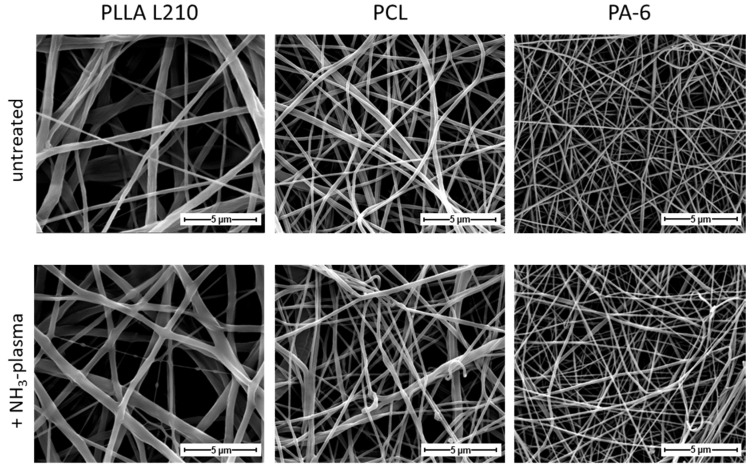
High-resolution scanning electron micrographs of morphology of untreated (**first row**) and NH_3_-plasma treated (**second row**) nanofibrous PLLA L210, PCL and PA-6 nonwovens (magnification ×8000, bar = 5 µm).

**Figure 4 materials-15-02014-f004:**
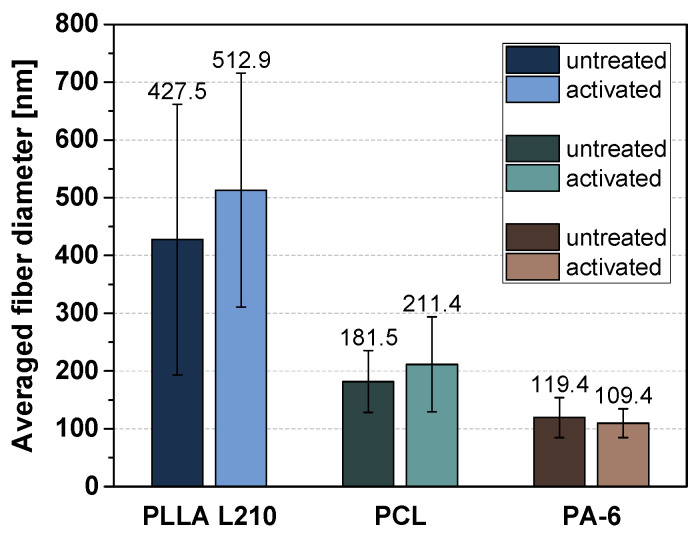
Fiber diameter of untreated and NH_3_-plasma functionalized PLLA L210, PCL and PA-6 nonwovens based on individual measurement points. For each polymer, 10 fibers were measured manually in 5 different SEM images (*n* = 50).

**Figure 5 materials-15-02014-f005:**
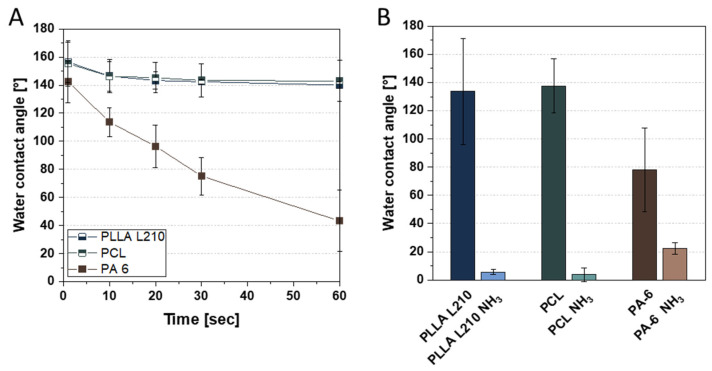
Water contact angle of (**A**) time-resolved measurement over 60 s using untreated nonwoven material and (**B**) measurement after a few seconds using untreated and NH_3_-plasma functionalized PLLA L210, PCL and PA-6 nonwovens.

**Figure 6 materials-15-02014-f006:**
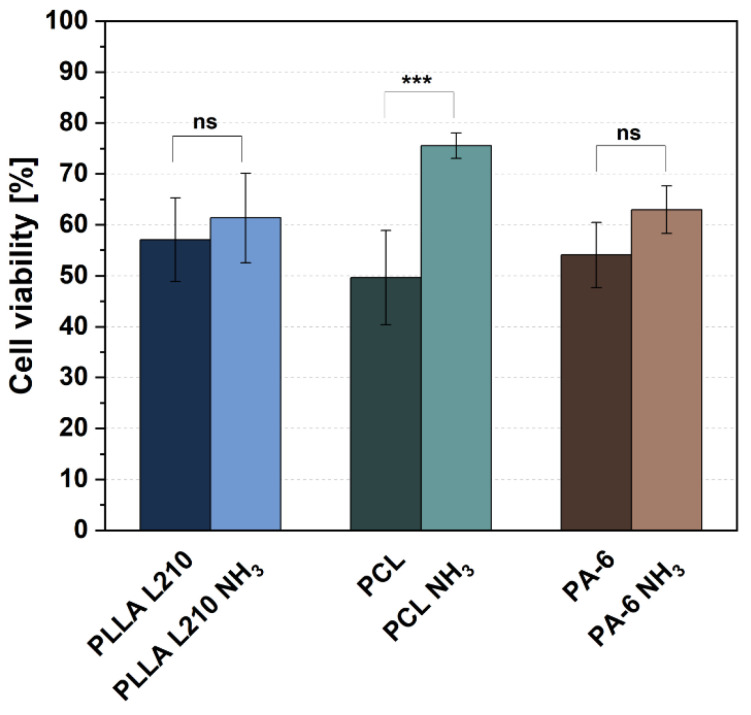
Relative viability of human endothelial EA.hy926 cells on unmodified and NH_3_-plasma functionalized polymeric nonwovens after 48 h (mean + SD, *n* = 6, one-way ANOVA, ns = not significant, and *** *p* < 0.001).

**Figure 7 materials-15-02014-f007:**
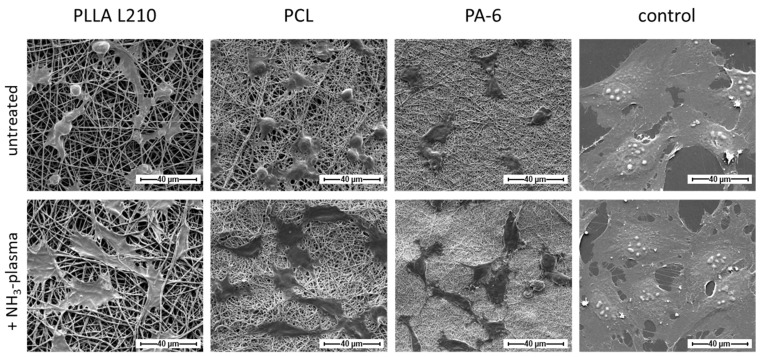
Cell morphology of human endothelial cells (EA.hy926) on untreated and NH3-plasma functionalized PLLA L210, PCL and PA-6 nonwovens and on a planar control surface (NC) after 48 h (SEM, bar = 40 µm).

**Figure 8 materials-15-02014-f008:**
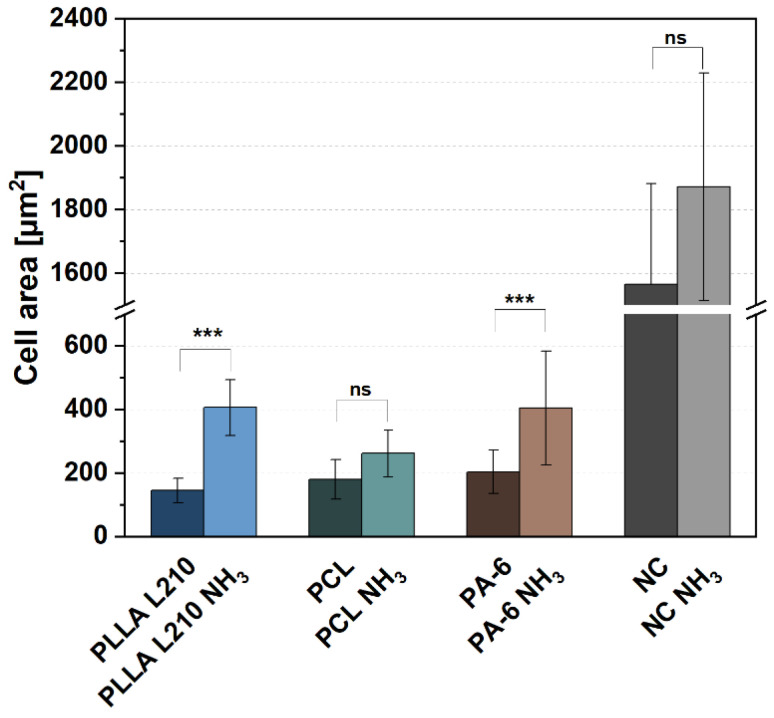
Spreading of human endothelial EA.hy926 cells on untreated and NH_3_-plasma functionalized polymeric nonwovens after 48 h (mean + SD, *n* = 40, one-way ANOVA, ns = not significant, and *** *p* < 0.001).

**Figure 9 materials-15-02014-f009:**
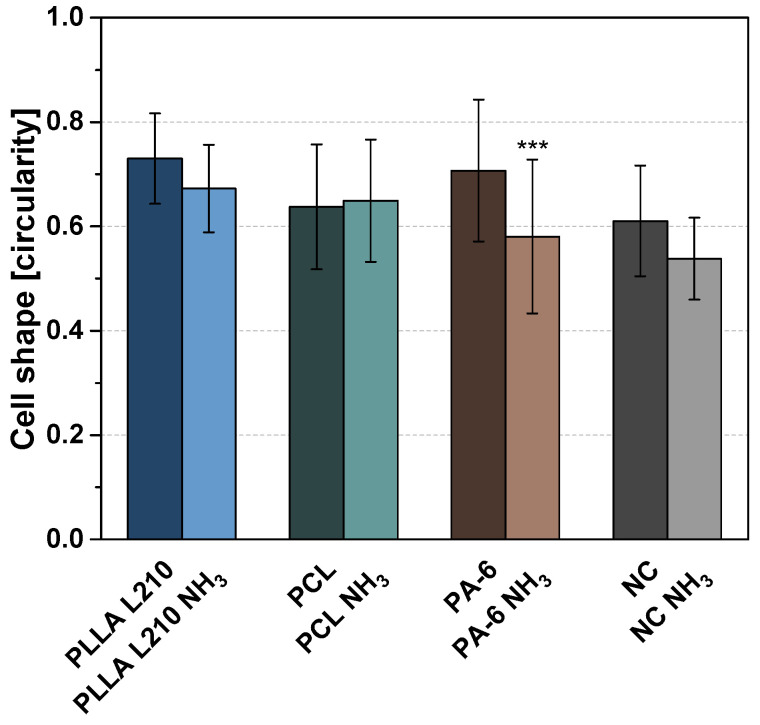
Cell shape described by cell circularity of human endothelial EA.hy926 cells on untreated and NH_3_-plasma functionalized polymeric nonwovens after 48 h (mean + SD, *n* = 40, one-way ANOVA, *** *p* < 0.001).

**Figure 10 materials-15-02014-f010:**
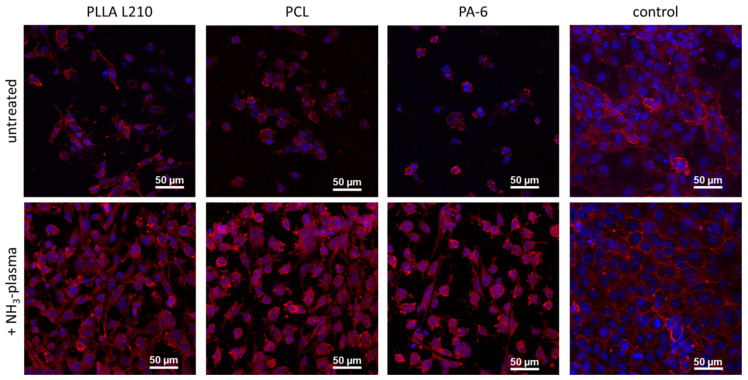
Fluorescent staining of F-actin in human endothelial EA.hy926 cells grown for 48 h on untreated and NH_3_-plasma functionalized polymeric nonwovens (red: phalloidin-TRITC for F-actin, blue: Hoechst-staining indicating cell nuclei, confocal microscopy, and bar = 50 μm).

**Figure 11 materials-15-02014-f011:**
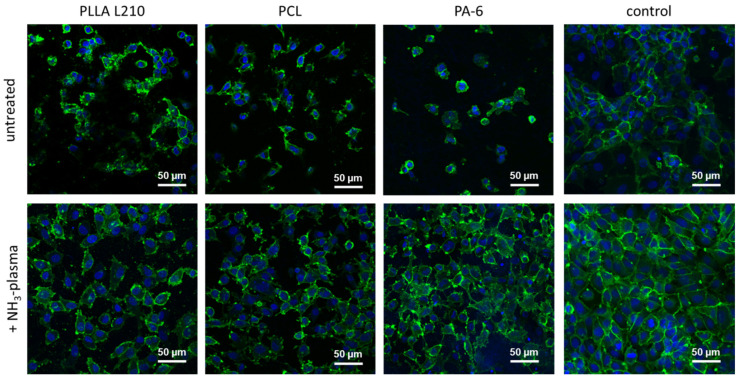
Immunostainings of CD31 in human endothelial EA.hy926 cells on unmodified and NH_3_-plasma functionalized polymeric nonwovens after 48 h (green: CD31, blue: Hoechst-staining indicating cell nuclei, confocal microscopy, and bar = 50 μm).

**Table 1 materials-15-02014-t001:** Summary of wettability and surface free energy of untreated and NH_3_-plasma functionalized PLLA L210, PCL and PA-6 nonwovens. MSA measurement after 2 s, *n* = 3.

	Untreated Nonwovens			NH_3_-Plasma-Activated Nonwovens		
	PLLA L210 *	PCL	PA-6	PLLA L210 *	PCL	PA-6
	Mean ± SD	Mean ± SD	Mean ± SD	Mean ± SD	Mean ± SD	Mean ± SD
Water [°]	133.6 ± 5.6	137.6 ± 3.8	78.0 ± 22.2	37.8 ± 1.7	19.3 ± 4.8	29.8 ± 4.1
Diiodomethane [°]	18.1 ± 2.4	21.6 ± 11.1	26.8 ± 7.6	64.6 ± 0.3	25.9 ± 1.0	31.6 ± 3.3
Surface free energy [mN/m]	57.1 ± 2.8	57.3 ± 5.3	48.5 ± 9.6	59.5 ± 1.3	75.9 ± 1.9	70.7 ± 3.5
Disperse fraction [mN/m]	48.3 ± 0.7	47.3 ± 3.5	45.5 ± 2.9	25.9 ± 0.2	45.8 ± 0.4	43.6 ± 1.4
Polar fraction [mN/m]	8.8 ± 2.1	10.0 ± 1.8	3.1 ± 6.8	33.6 ± 1.0	30.1 ± 1.6	27.1 ± 2.0

* washed.

**Table 2 materials-15-02014-t002:** Exemplary potentials and limitations of plasma surface treatment of electrospun polymer scaffolds.

	Control of	Selected Parameters	Selected Limitations
Plasma →← Electrospinning	Fiber topography	Adjustable fiber diameters ranging from nano- to micrometers	No electrospinning of chemical inert polymers, complex 3D scaffolds
	Biological response	Biofilm inhibition and endothelial cell enhancement	Polymer specific nonwovens, long-term stability of plasma activation
	Tribological properties	Surface roughness, sliding vs. adhesive	Specific hydrophilicity
	Local- or side-specific modifications	Layer structure, composites, and top vs. bottom	Possible material defects, e.g., delamination, etching or radiation
	Patient individuality	Easy and fast plasma modification, e.g., during interventions	Transferability, e.g., no defined electrospinning and plasma parameters
	Surface chemistry	Specific functional groups for graftings, coatings or drugs	Material and plasma dependence

## Data Availability

The data presented in this study are available upon request from the corresponding author.

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
