# Peer review of "Accelerated Endothelialization of Nanofibrous Scaffolds for Biomimetic Cardiovascular Implants"

_materials, 2022, doi:10.3390/ma15062014_

Round 1

Reviewer 1 Report

The work submitted for review and entitled "Accelerated endothelialization of nanofibrous scaffolds for biomimetic cardiovascular implants" concerns the interesting issue of the applicability of biomaterials. The topic is important and attractive. It enriches the current state of knowledge in this field. Some concerns and tips are presented below/;

  1. The abstract is correctly written and allows the reader to get acquainted with the research topic. I have no major objections to this part.
  2. The introduction part is fine, however, part of the introduction lines 33-105 contain facts and important information that is not supported by any literature references. I suggest refining the references and supplementing them.
  3.  Apart from synthetic biopolymers, also natural biopolymers have biostatic properties. I think that a mention of such solutions and a possible discussion of pros and cons would enrich this part of the literature.
  4. The experimental part is fine. The reviewer has no major objections to this part.
  5. The morphology of polymeric nonwovens presented in section 3.1 shows the homogeneous surface of the fibers and their good development in the preparation process. However, they do not reveal any morphological changes before and after the plasma treatment process. Can the authors comment on it during actual literature? Lack such references in the discussion part (part 4.1). Maybe it is worth presenting the photo in higher magnification or trying, for example, with the AFM technique of the fiber surface. I know from experience that it can be valuable.
  6. In general, I suggest that you complete references throughout the work. 
  7. The presented paper is well written and easy to read.

Reviewer 2 Report

This manuscript by Claudia Matschegewski et al. investigated the endothelialization performance on three types of polymeric nanofiber nonwovens (PLLA L210, PCL, and PA-6) before and after plasma treatment. They systematically characterized the nonwovens by SEM, WCA, surface energy, etc and evaluated adhered cell morphology, endothelialization, and CD31 expressions on the surfaces of the nonwovens. Overall, the structure of this work is well designed, and this reviewer does not have any major technically related issues. Before it may be accepted, there are some concerns should be addressed as follows:

  1. Please put the Figure 1 into supporting materials as it is a summary of previous work. Also, revise the typo ‘biopint’ to ‘bioprint’ in Figure 1 caption.
  2. The authors should introduce more details on the benefits of NH3-plasma treatment for surface biofunctionalization because it is the core technique involved in this study.
  3. ‘Surface plasma modification of nonwovens’, the authors only used one condition ‘1 minute and 60% generator output’ for the surface treatment”? Any optimization results or other references for this plasma activation process?
  4. The cell viability on the samples after the NH3-plasma treatment was low (60%-80%), although they increased compared with the control samples. I am wondering if this low cell viability would support the cell endothelialization in a long term. Please comment on that.
  5. Please quantify the actin formation and CD31 expression results in the Figures 9 and 10, rather than just showing the fluorescence images. A recent work also related to the protein/cell-biomaterial interactions would be helpful for this (Chem. Mater. 2018, 30, 4372).

Minor:

  1. It is hard to distinguish the WCA curves of PLLA L210 and PCL in Figure 4A. Please re-design the Figure.

Reviewer 3 Report

Dear Authors,
Thank you for very interesting paper in the area of materials for biomimetic cardiovascular implants.

I have some comments and suggestions as below.

  1. Materials and methods section - Line 146. What is the unit of [v%]?
  2. Materials and methods section - Line 157. What was the pressure in the chamber (vacuum level)?
  3. Materials and methods section - Line 162. What device was used to depostion of gold coating on the samples (deposition conditions)?
  4. Paragraph 2.2 - What modes of SEM observation were used?
  5. Par. 2.3. - Why did you used the surface plasma modification process? What is the aim of this process in this case? Please, explain it.
  6. Par. 2.7 - Why did you used CO2 to drying process? Please, explain it.
  7. Results - Figure 4 - Error in the axis description ("angel" instead "angle").

Thank you for all answers in advance.
Best regards,
Reviewer

Reviewer 4 Report

The manuscript “Accelerated endothelialization of nanofibrous scaffolds for biomimetic cardiovascular implants” (materials-1605900) by Matschegewski et al. examined the potential of electrospun fibrous biomaterials to serve as engineered biocompatible and biomimetic cardiac scaffolds. The topic is interesting, but I think this article should reconsider after proper changes in major revision.

  1. I would encourage and advise the authors to adopt some of the additional references published by MDPI in the introduction section:

Tresca Stress Simulation of Metal-on-Metal Total Hip Arthroplasty during Normal Walking Activity. Materials (Basel). 2021, 14, 7554. https://doi.org/10.3390/ma14247554

The Effect of Bottom Profile Dimples on the Femoral Head on Wear in Metal-on-Metal Total Hip Arthroplasty. J. Funct. Biomater. 2021, 12, 38. https://doi.org/10.3390/jfb12020038

  1. To improve the quality of English used in this manuscript and make sure English language, grammar, punctuation, spelling, and overall style are correct, further proofreading is needed. As an alternative, the authors can use the MDPI English proofreading service for this issue.
  2. The author seems to have made an error in using uppercase and lowercase in the title of the present article (line 2-3) and the subsection that should be corrected.
  3. In the abstract section (line 12-28) the authors needs to add qualitative result rather than the only qualitative result as presented in the manuscript.
  4. In the keyword section (line 29-30), the authors are recommended only put a maximum 5 (five) keywords, 7 (seven) as presented in the manuscript is too much.
  5. The introduction (line 32-137) is too long and not clearly highlight the state of the art and research novelty in the present manuscript. The authors should shorten the introduction and highlight it more clearly state of the art and research novelty.
  6. In Figure 1 at the introduction section (line 78-82), why do the authors use only web of science database for the present annual number of publications that related to conducted study? For a more comprehensive explanation, it would be interesting to include two other main database, Scopus and PubMed.
  7. In the materials and methods section (line 138-252) the authors should add one systematic figure to illustrate the workflow of experimental testing in the present study to make the reader easier to understand rather than only using dominant text to explain.
  8. The author must provide a detailed specification and use condition more detail regarding all tools used in the research carried out so that the reader can estimate the accuracy and differences in the results that the authors describe due to the use of different tools in future studies.
  9. In the results section, authors are advised to compare the results they obtain with previous similar/identical studies if it is possible after line 427.
  10. To improve the presented data in the present manuscript, Figure 3 (line 269-271), Figure 4 (line 277-279), Figure 5 (line 310-312), Figure 7 (line 355-357), and Figure 8 (line 382-385) should use colored chart rather than only black and white.
  11. In the last paragraph of the discussion section, the authors should add one paragraph about the limitations of the research conducted after line 586.
  12. Further research needs to be explained in the conclusion section.
  13. Please make sure the authors have used the Materials, MDPI format correctly. The authors can download published manuscripts by Materials, MDPI, and compare them with the present author's manuscript to ensure typesetting is appropriate.

Round 2

Reviewer 2 Report

The authors have fully addressed my comments, this reviewer does not have further concerns.

Reviewer 4 Report

Dear Matschegewski et al.,

After carefully reading the author's revised manuscript entitled "Accelerated endothelialization of nanofibrous scaffolds for biomimetic cardiovascular implants" (materials-1605900) by Matschegewski et al., The authors have been made significant improvements in the revised manuscript. Also, all of the issue in my review report have been addressed precisely.

With my pleasure, I recommend the manuscript should be accepted for publication on Materials.

Best regards,

The Reviewer